# Projection-based Lyapunov method for fully heterogeneous weakly-coupled MDPs

Xiangcheng Zhang[1][*][†]   Yige Hong[2][*]   Weina Wang[2]

[1]Weiyang College, Tsinghua University
[2] Computer Science Department, Carnegie Mellon University
xc-zhang21@mails.tsinghua.edu.cn
{yigeh,weinaw}@cs.cmu.edu

## Abstract

Heterogeneity poses a fundamental challenge for many real-world large-scale decision-making problems but remains largely understudied. In this paper, we study the *fully heterogeneous* setting of a prominent class of such problems, known as weakly-coupled Markov decision processes (WCMDPs). Each WCMDP consists of $N$ arms (or subproblems), which have distinct model parameters in the fully heterogeneous setting, leading to the curse of dimensionality when $N$ is large. We show that, under mild assumptions, an efficiently computable policy achieves an $O(1/\sqrt{N})$ optimality gap in the long-run average reward per arm for fully heterogeneous WCMDPs as $N$ becomes large. This is the *first asymptotic optimality result* for fully heterogeneous average-reward WCMDPs. Our main technical innovation is the construction of projection-based Lyapunov functions that certify the convergence of rewards and costs to an optimal region, even under full heterogeneity.[1]

## 1 Introduction

Heterogeneity poses a fundamental challenge for many real-world decision-making problems, where each problem consists of a large number of interacting components. However, despite its practical significance, heterogeneity remains largely understudied in the literature. In this paper, we study *heterogeneous* settings of a prominent class of such problems, known as weakly-coupled Markov decision processes (WCMDPs) [23]. A WCMDP consists of $N$ *arms* (or *subproblems*), where each arm itself is a Markov decision process (MDP). In a heterogeneous setting, the MDPs could be distinct. At each time step, the decision-maker selects an action for each arm, which affects the arm's transition probabiltities and reward, and then the arms make state transitions independently. However, these actions are subject to a set of global *budget constraints*, where each constraint limits one type of total cost across all arms at each time step. The objective is to find a policy that maximizes the long-run *average reward* over an infinite time horizon. We focus on the *planning* setting, where all the model parameters (reward function, cost functions, budget, and transition kernel) are known.

WCMDPs have been used to model a wide range of applications, including online advertising [7, 57], job scheduling [48], healthcare [5], surveillance [40], and machine maintenance [21]. A faithful modeling of these applications calls for *heterogeneity*. For instance, in [5], arms are beneficiaries of a health program and they could react to interventions differently; in [40], arms are targets of

---

[*]These authors contributed equally to this work.

[†]Work done during a visit at Carnegie Mellon University.

[1]A full version of this paper [54] can be found at https://www.arxiv.org/abs/2502.06072v6

39th Conference on Neural Information Processing Systems (NeurIPS 2025).

surveillance who have different locations and probabilities to be exposed; in [21], arms are machines that could require distinct repair schedules.

Although heterogeneity is crucial in the modeling of these applications, most existing work on average-reward WCMDPs establishes asymptotic optimality only for the homogeneous setting where all arms share the same set of model parameters [14, 15, 22, 24–27, 39, 42, 47]. Only a few exceptions [25, 39, 46] address heterogeneity, but in highly specialized settings. Among these, a common approach to handle heterogeneity is to consider the *typed heterogeneous* setting, where the $N$ arms are divided into a constant number of types as $N$ scales up, with each type having distinct model parameters. While heterogeneous WCMDPs have been studied under the finite-horizon total-reward and discounted-reward criteria, these results do not extend to the average-reward setting we consider. We review related work in more detail at the end of this section and also in Appendix A.

The key distinction between the homogeneous (or typed heterogeneous) setting and the fully heterogeneous setting is whether the arms can be divided into a *constant* number of homogeneous groups. In the former, the system dynamics depends only on the fraction of arms in each state in each homogeneous group. Thus, the effective dimension of the state space is *polynomial* in $N$. In contrast, in the fully heterogeneous setting, the state space grows *exponentially* in $N$, making the problem truly high-dimensional.

**Our contribution.**  In this paper, we study *fully heterogeneous* WCMDPs. We propose a policy we call the *ID policy with reassignment*, which generalizes the ID policy in the literature [27]. In our policy, we first perform an ID reassignment algorithm to reorder the arms, which ensures proper arm prioritization during policy execution. We then run a variant of the ID policy adapted to handle heterogeneity, which consists of two phases. The first phase is a pre-processing phase, where we compute an *optimal single-armed policy* for each arm (denoted as $\bar{\pi}_i^*$ for the $i$-th arm) that prescribes the *ideal action* the arm would take at each state. The second phase is the real-time phase. At each time step, the policy iterates over the arms according to their reassigned IDs, and it lets as many arms as possible follow their respective ideal actions while satisfying the budget constraints. Unlike the original ID policy, which has only one optimal single-armed policy due to homogeneous arms, our policy computes $N$ optimal single-armed policies, one for each arm. Our proposed policy is efficiently computable, with computational complexity polynomial in $N$.

We prove that the proposed ID policy with reassignment achieves an $O(1/\sqrt{N})$ optimality gap under mild assumptions as the number of arms $N$ becomes large. Here, the optimality gap refers to the difference between the long-run average reward per arm under our policy and that under the optimal policy. *This is the first result establishing asymptotic optimality for fully heterogeneous average-reward WCMDPs.*

We remark that the original ID policy was designed for a special case of WCMDPs known as restless bandits, and it is for the homogeneous setting. While the generalization in our proposed policy is natural, identifying the appropriate generalization and establishing its optimality gap in the fully heterogeneous setting are technically challenging and require new theoretical approaches.

**Technical novelty.**  The main technical innovation of the paper is the introduction of a novel Lyapunov function for fully heterogeneous WCMDPs. Specifically, to prove the asymptotic optimality of a policy, a key step is to show that the system state is globally attracted to an *optimal region* where most arms can follow the ideal actions generated by their respective optimal single-armed policies $\bar{\pi}_i^*$'s. In the homogeneous setting, we can prove such convergence using state aggregation techniques that rely on the symmetry of arms. However, in the heterogeneous setting, states of different arms cannot be aggregated since arms are no longer symmetric. Our technique is to *project* arm states onto a set of carefully selected feature vectors, and define the Lyapunov function based on these projections. These feature vectors encode the minimal amount of information needed to evaluate the relevant functions of the system state (e.g., instantaneous reward or cost) and predict their future expectations. This projection-based Lyapunov function provides a principled way to measure deviations of the system state from the optimal region in a fully heterogeneous setting. A more detailed discussion of this approach can be found in Section 4.

Beyond WCMDPs, our techniques have the potential to be applied to more general heterogeneous large stochastic systems. Heterogeneity has been a topic of strong interest in these systems, but it is known to be a challenging problem with limited theoretical results. Only recently have there been

notable breakthroughs. [1, 2] extended the popular mean-field analysis to a class of heterogeneous large stochastic systems for the first time, but the results are only for transient distributions. Another line of work [55, 56] studied heterogeneous load-balancing systems. They first analyzed the transient distributions and then used interchange-of-limits arguments to extend the results to steady state. Our method provides a more direct framework for steady-state analysis and has the potential to generalize to a broader range of heterogeneous stochastic systems.

**Related work.** WCMDPs have been extensively studied with a rich body of literature. Here we briefly overview the most relevant work and refer the reader to Appendix A for a detailed survey.

We first focus on the *average-reward* criterion. As mentioned earlier, most existing work considers the *homogeneous setting*. Early work on WCMDPs primarily focuses on a special case known as the *restless bandit (RB)* problem, where each arm's MDP has a binary action space (active and passive actions) and there is only one budget constraint that limits the total number of active actions across all arms at each time step. The seminal work by Whittle [44] introduced the RB problem and the celebrated Whittle index policy, which was later shown to achieve an $o(1)$ optimality gap as $N \rightarrow \infty$ under a set of conditions [42]. Subsequent work on RBs has focused on designing policies that achieve asymptotic optimality under more relaxed conditions [25, 27, 39, 47], as well as improving the optimality gap to $O(1/\sqrt{N})$ [25, 27] or $O(\exp(-cN))$ [14, 15, 26]. Among these papers, [25, 39] address heterogeneous RBs. However, [39] focuses on the *typed heterogeneous* setting, where the $N$ arms are divided into a constant number of types as $N \rightarrow \infty$. The paper [25] includes an extension to the fully heterogeneous setting. However, for their result to yield asymptotic optimality, there need to be further assumptions on the orders of the so-called synchronization times in the paper. The policies in both papers cannot be straightforwardly extended to general WCMDPs, which have multiple actions, multiple budget constraints, and state-dependent cost functions.

Beyond RBs, work on general average-reward WCMDPs is scarce. The closest to ours are [24, 39, 46], which studied WCMDPs with a single budget constraint and established $o(1)$ optimality gaps, but again in the homogeneous setting [24] or the *typed heterogeneous* setting [39, 46]. More recently, [22] proved the first $o(1)$ optimality gap result for WCMDPs with general budget constraints, but in the *homogeneous* setting.

Under the *finite-horizon total-reward* or *discounted-reward* criteria, there has been more work on heterogeneous settings, including both the typed heterogeneous setting [12, 17] and, more recently, the fully heterogeneous setting [8–10, 50]. However, the optimality gap in these papers generally grows *super-linearly* with the (effective) time horizon, except under restrictive conditions. Consequently, it is difficult to extend these results to the average-reward setting and still achieve asymptotic optimality.

Although our work focuses on the planning setting where all model parameters are known, there has been growing interest in developing reinforcement learning algorithms for the learning setting with unknown parameters [3, 6, 13, 29, 30, 32, 33, 36, 45, 46]. Many of these approaches rely on well-designed planning policies as a foundation to achieve learning efficiency. In this context, our results can serve as an important building block for developing model-based learning algorithms for fully heterogeneous WCMDPs.

**General notation.** Let $\mathbb{R}$, $\mathbb{N}$, and $\mathbb{N}_+$ denote the sets of real numbers, nonnegative integers, and positive integers, respectively. Let $[N] \triangleq \{1, 2, \ldots, N\}$ for any $N \in \mathbb{N}_+$ and $[n_1 : n_2] \triangleq \{n_1, n_1 + 1, \ldots, n_2\}$ for $n_1, n_2 \in \mathbb{N}_+$ with $n_1 \leq n_2$. Let $[0, 1]_N = \{i/N : i \in \mathbb{N}, 0 \leq i/N \leq 1\}$, the set of integer multiples of $1/N$ in $[0, 1]$. For a matrix $A \in \mathbb{R}^{d \times d}$, we denote its operator norm as $\|A\|_p = \sup_{x \neq \mathbf{0}} \|Ax\|_p / \|x\|_p$. We use boldface letters to denote matrices, and regular letters to denote vectors and scalars. We write $\mathbb{R}^{\mathbb{S}}$ for the set of real-valued vectors indexed by elements of $\mathbb{S}$, or equivalently, the set of real-valued functions on $\mathbb{S}$; for each $v \in \mathbb{R}^{\mathbb{S}}$, let $v(s)$ to denote its element corresponding to $s \in \mathbb{S}$.

## 2 Problem setup

We consider a weakly-coupled Markov decision process (WCMDP) that consists of $N$ arms. Each arm has an ID $i \in [N]$ and is associated with a smaller MDP denoted as $\mathcal{M}_i = \left(\mathbb{S}, \mathbb{A}, \mathbb{P}_i, r_i, (c_{k,i})_{k \in [K]}\right)$. Here $\mathbb{S}$ and $\mathbb{A}$ are the state space and the action space, respectively, both assumed to be finite; $\mathbb{P}_i$ describes the transition probabilities with $\mathbb{P}_i(s' \mid s, a)$ being the transition probability from state $s$

to state $s'$ when action $a$ is taken. The state transitions of different arms are independent given the actions. When arm $i$ is in state $s$ and we take action $a$, a reward $r_i(s, a)$ is generated, as well as $K$ types of costs $c_{k,i}(s, a), k \in [K]$. We assume that the costs are nonnegative, i.e., $c_{k,i}(s, a) \geq 0$ for all $i \in \mathbb{N}_+, k \in [K], s \in \mathbb{S}$, and $a \in \mathbb{A}$. Note that we allow the arms to be *fully heterogeneous*, i.e., the $\mathcal{M}_i$'s can be *all distinct*.

When taking an action for each arm in this $N$-armed system, we are subject to budget constraints. Specifically, suppose each arm $i$ is in state $s_i$. Then the actions, $a_i$'s, should satisfy the constraints:

$$\sum_{i \in [N]} c_{k,i}(s_i, a_i) \leq \alpha_k N, \quad \forall k \in [K], \tag{1}$$

where each $\alpha_k > 0$ is a constant independent of $N$, and $\alpha_k N$ is referred to as the *budget* for type-$k$ cost. We assume that there exists an action $0 \in \mathbb{A}$ that does not incur any type of cost for any arm at any state, i.e., $c_{k,i}(s, 0) = 0$ for all $k \in [K], i \in [N], s \in \mathbb{S}$. This assumption guarantees that there always exist valid actions (e.g., taking action 0 for every arm) regardless of the states of the arms.

**Policy and system state.** A policy $\pi$ for the $N$-armed problem specifies the action for each of the $N$ arms, in a possibly history-dependent way. Under policy $\pi$, let $S_{i,t}^\pi$ denote the state of the $i$th arm at time $t$, and we refer to $\boldsymbol{S}_t^\pi \triangleq (S_{i,t}^\pi)_{i \in [N]}$ as the *system state*. Similarly, let $A_{i,t}^\pi$ denote the action applied to arm $i$ at time $t$, and we refer to $\boldsymbol{A}_t^\pi \triangleq (A_{i,t}^\pi)_{i \in [N]}$ as the *system action*. In this paper, we also use an alternative representation of the system state, denoted as $\boldsymbol{X}_t^\pi$ and defined as follows. Let $X_{i,t}^\pi = (X_{i,t}^\pi(s))_{s \in \mathbb{S}} \in \mathbb{R}^{|\mathbb{S}|}$ be a row vector where the entry corresponding to state $s$ is given by $X_{i,t}^\pi(s) = \mathbb{1}\{S_{i,t}^\pi = s\}$; i.e., $X_{i,t}^\pi$ is a one-hot row vector whose $s$'s entry is 1 if $S_{i,t}^\pi = s$ and is 0 otherwise. Then let $\boldsymbol{X}_t^\pi$ be an $N \times |\mathbb{S}|$ matrix whose $i$th row is $X_{i,t}^\pi$. It is easy to see that $\boldsymbol{X}_t^\pi$ contains the same information as $\boldsymbol{S}_t^\pi$, and we refer to both of them as the system state. In this paper, we often encounter vectors like $X_{i,t}^\pi = (X_{i,t}^\pi(s))_{s \in \mathbb{S}}$, whose entries correspond to different states in $\mathbb{S}$. For such vectors, say $u$ and $v$, we use the inner product to write a sum for convenience $\langle u, v \rangle \triangleq \sum_{s \in \mathbb{S}} u(s)v(s)$. We sometimes omit the superscript $\pi$ when it is clear from context.

**Maximizing average reward.** Our objective is to maximize the long-run time-average reward subject to the budget constraints. To be more precise, we follow the treatment for maximizing average reward in [35]. For any policy $\pi$ and an initial state $\boldsymbol{S}_0$ of the $N$-armed system, consider the *limsup* average reward $R^+(\pi, \boldsymbol{S}_0)$ and the *liminf* average $R^-(\pi, \boldsymbol{S}_0)$, defined as $R^+(\pi, \boldsymbol{S}_0) = \limsup_{T \to \infty} \frac{1}{T} \sum_{t=0}^{T-1} \frac{1}{N} \sum_{i \in [N]} \mathbb{E}\left[r_i(S_{i,t}^\pi, A_{i,t}^\pi)\right]$ and $R^-(\pi, \boldsymbol{S}_0) = \liminf_{T \to \infty} \frac{1}{T} \sum_{t=0}^{T-1} \frac{1}{N} \sum_{i \in [N]} \mathbb{E}\left[r_i(S_{i,t}^\pi, A_{i,t}^\pi)\right]$. If $R^+(\pi, \boldsymbol{S}_0) = R^-(\pi, \boldsymbol{S}_0)$, then the average reward of policy $\pi$ under the initial condition $\boldsymbol{S}_0$ exists and is defined as

$$R(\pi, \boldsymbol{S}_0) = R^+(\pi, \boldsymbol{S}_0) = R^-(\pi, \boldsymbol{S}_0) = \lim_{T \to \infty} \frac{1}{T} \sum_{t=0}^{T-1} \frac{1}{N} \sum_{i \in [N]} \mathbb{E}\left[r_i(S_{i,t}^\pi, A_{i,t}^\pi)\right]. \tag{2}$$

Note that these reward notions divide the total reward from all arms by the number of arms, $N$, measuring the reward *per arm*. The WCMDP problem is to solve the following optimization problem:

$$\underset{\text{policy } \pi}{\text{maximize}} \quad R^-(\pi, \boldsymbol{S}_0) \tag{3a}$$

$$\text{subject to} \quad \sum_{i \in [N]} c_{k,i}(S_{i,t}^\pi, A_{i,t}^\pi) \leq \alpha_k N, \quad \forall k \in [K], \forall t \geq 0. \tag{3b}$$

Let the optimal value of this problem be denoted as $R^*(N, \boldsymbol{S}_0)$. Note that since the WCMDP is an MDP with finite state and action space, if we replace the $R^-(\pi, \boldsymbol{S}_0)$ in the objective (3a) with $R^+(\pi, \boldsymbol{S}_0)$, the optimal value stays the same [35, Proposition 9.1.6].

**Asymptotic optimality.** Recall that exactly solving the WCMDP problem is PSPACE-hard [34]. In this paper, our goal is to design a policy $\pi$ that is *efficiently computable* and *asymptotically optimal* as $N \to \infty$, with the following notion for asymptotic optimality. For any policy $\pi$, we define its *optimality gap* as $R^*(N, \boldsymbol{S}_0) - R^-(\pi, \boldsymbol{S}_0)$. We say the policy $\pi$ is *asymptotically optimal* if

$$R^*(N, \boldsymbol{S}_0) - R^-(\pi, \boldsymbol{S}_0) = o(1) \quad \text{as} \quad N \to \infty. \tag{4}$$

When we take this asymptotic regime as $N \to \infty$, we keep the number of constraints, $K$, as well as the budget coefficients, $\alpha_1, \alpha_2, \ldots, \alpha_K$, fixed. We assume that the reward functions and cost functions are uniformly bounded, i.e., $\sup_{i \in \mathbb{N}_+} \max_{s \in \mathbb{S}, a \in \mathbb{A}} |r_i(s, a)| \triangleq r_{\max} < \infty$ and $\sup_{i \in \mathbb{N}_+} \max_{k \in [K], s \in \mathbb{S}, a \in \mathbb{A}} c_{k,i}(s, a) \triangleq c_{\max} < \infty$. This notion for asymptotic optimality is consistent with that in the existing literature (e.g., [39, Definition 4.11]). We are interested in not only achieving asymptotic optimality but also characterizing the *order* of the optimality gap.

In the remainder of this paper, we focus on stationary Markov policies, which are sufficient for achieving the optimal value because the WCMDP problem is an MDP with finite state and action spaces [35, Theorem 9.1.8]. Under any stationary Markovian policy, the long-run reward $R(\pi, \boldsymbol{S}_0) = R^+(\pi, \boldsymbol{S}_0) = R^-(\pi, \boldsymbol{S}_0)$ is always well-defined [35, Theorem 8.1.1].

**LP relaxation and an upper bound on optimality gap.** We consider the linear program (LP) below, which will play a critical role in performance analysis and policy design:

$$R_N^{\text{rel}} \triangleq \operatorname*{maximize}_{(y_i(s,a))_{i \in [N], s \in \mathbb{S}, a \in \mathbb{A}}} \frac{1}{N} \sum_{i \in [N]} \sum_{s \in \mathbb{S}, a \in \mathbb{A}} y_i(s, a) r_i(s, a) \tag{5a}$$

$$\text{subject to} \quad \frac{1}{N} \sum_{i \in [N]} \sum_{s \in \mathbb{S}, a \in \mathbb{A}} y_i(s, a) c_{k,i}(s, a) \leq \alpha_k, \; \forall k \in [K], \tag{5b}$$

$$\sum_{s' \in \mathbb{S}, a' \in \mathbb{A}} \mathbb{P}_i(s \mid s', a') y_i(s', a') = \sum_{a \in \mathbb{A}} y_i(s, a), \; \forall s \in \mathbb{S}, \forall i \in [N], \tag{5c}$$

$$\sum_{s' \in \mathbb{S}, a' \in \mathbb{A}} y_i(s', a') = 1, \; y_i(s, a) \geq 0, \; \forall s \in \mathbb{S}, \forall a \in \mathbb{A}, \forall i \in [N]. \tag{5d}$$

Lemma 1 below establishes a connection between this LP and the WCMDP.

**Lemma 1.** *The optimal value of any $N$-armed WCMDP problem is upper bounded by the optimal value of the corresponding linear program in* (5), *i.e.,*

$$R^*(N, \boldsymbol{S}_0) \leq R_N^{\text{rel}}, \quad \forall N, \forall \boldsymbol{S}_0.$$

An immediate implication of Lemma 1 is that for any policy $\pi$, its optimality gap is upper bounded as

$$R^*(N, \boldsymbol{S}_0) - R^-(\pi, \boldsymbol{S}_0) \leq R_N^{\text{rel}} - R^-(\pi, \boldsymbol{S}_0). \tag{6}$$

Therefore, to derive an upper bound for the optimality gap, it suffices to control $R_N^{\text{rel}} - R^-(\pi, \boldsymbol{S}_0)$, which we will show is $O(1/\sqrt{N})$ in Theorem 1.

To see the intuition of Lemma 1, we interpret the optimization variable $y_i(s, a)$ as the long-run fraction of time arm $i$ spends in state $s$ and takes action $a$. We refer to $y_i(s, a)$ as arm $i$'s *state-action frequency* for the state-action pair $(s, a)$. Then the constraints in (5b) of the LP can be viewed as relaxations of the budget constraints in (3b) for the WCMDP. The constraints in (5c)–(5d) guarantee that $y_i(s, a)$'s are proper stationary time fractions. Therefore, the LP is a relaxation of the WCMDP and thus achieves a higher optimal value. The proof of Lemma 1 is provided in Appendix C of [54].

Our LP (5) serves a similar role to the LP used in previous work on restless bandits and WCMDPs (see, e.g., [15, 22, 24, 25, 39, 42, 46]), but with different forms and dependencies on $N$. Both our LP and the LP in previous work relax the hard budget constraints to time-average constraints. However, in the homogeneous arm setting [15, 22, 24, 42], the LP has only one set of state-action frequencies $y(s, a)$, and the LP is independent of $N$. As a result, both the optimal value of the LP and the complexity of solving it are independent of $N$. Prior work for the typed-heterogeneous setting [25, 39, 46] divides the arms into a constant number $K$ of types and defines a set of state-action frequency $y_k(s, a)$ for each type $k$. The optimal value of the resulting LP and the complexity of solving it depend on the number of types $K$. Hong et al. [25] includes a generalization to the fully heterogeneous setting, resulting in an LP similar to ours, although restricted to restless bandits.

In our LP, we define a separate set of state-action frequencies $y_i(s, a)$ for each arm $i \in [N]$, making the LP explicitly depend on $N$. Therefore, the optimal value $R_N^{\text{rel}}$ depends on $N$, and the complexity of solving the LP grows with $N$. Nevertheless, because the number of variables and constraints scales linearly with $N$, our LP can still be solved in polynomial time.

---

**Algorithm 1** ID reassignment

---

1: **Input:** optimal state-action frequencies $(y_i^*(s,a))_{i\in[N],s\in\mathbb{S},a\in\mathbb{A}}$, budgets $(\alpha_k)_{k\in[K]}$
2: **Output:** new arm ID, recorded in newID($i$), for each arm with old ID $i \in [N]$
3: Compute $(C_{k,i}^*)_{i\in[N],k\in[K]}$ and the set of active constraints $\mathcal{A}$ using (8)
4: **if** $\mathcal{A} = \emptyset$ **then**
5:    newID($i$) $\leftarrow i$ for all $i \in [N]$           ▷ *No need for ID reassignment*
6: **else**
7:    Initialize $\mathcal{F} \leftarrow \emptyset$           ▷ *Set of arms that have been assigned new IDs*
8:    Initialize $\mathcal{D}_k \leftarrow \{i \in [N]\colon C_{k,i}^* \geq \delta\}$ for all $k \in \mathcal{A}$
9:    $\delta \leftarrow \alpha_{\min}/4 \triangleq \min_{k\in[K]} \alpha_k/4$;   $d \leftarrow \left\lceil \frac{(c_{\max}-\delta)K}{\alpha_{\min}/2-\delta} \right\rceil$
10:    **for** $\ell = 0, 1, \ldots, \lfloor N/d \rfloor - 1$ **do**
11:       $\mathcal{I}(\ell) \leftarrow [\ell d + 1 : (\ell+1)d]$;   $j \leftarrow \ell d + 1$
12:       **for** $k \in \mathcal{A}$ **do**
13:          **if** $\sum_{i\in\mathcal{F}} C_{k,i}^* \mathbb{1}\{\text{newID}(i) \in \mathcal{I}(\ell)\} < \delta$ **then**
14:             Pick any $i$ from $\mathcal{D}_k$ and set newID($i$) $\leftarrow j$; remove $i$ from $\mathcal{D}_{k'}$ for all $k'$; add $i$ to $\mathcal{F}$
15:             $j \leftarrow j + 1$
16:    For all $i \in [N]\setminus\mathcal{F}$, assign values to their newID($i$)'s randomly from $[N]\setminus\{\text{newID}(i')\colon i' \in \mathcal{F}\}$

---

## 3   ID policy with reassignment

In this section, we introduce the ID policy with reassignment, generalized from the ID policy designed for homogeneous restless bandits in the literature [27]. Our policy first performs an ID reassignment procedure, and then proceeds to run a variant of the ID policy adapted to handle heterogeneity. We begin by introducing a building block of our policy, referred to as optimal single-armed policies, followed by the ID reassignment algorithm and the execution of the adapted ID policy.

**Optimal single-armed policies.**   Once we obtain a solution to the LP in (5), we can construct a policy for each arm $i$, which we refer to as an *optimal single-armed policy* for arm $i$. In particular, let $(y_i^*(s,a))_{i\in[N],s\in\mathbb{S},a\in\mathbb{A}}$ be an arbitrary optimal solution to the LP in (5). Then for arm $i$, the optimal single-armed policy, $\bar{\pi}_i^*$, is defined as

$$\bar{\pi}_i^*(a \mid s) = \begin{cases} \frac{y_i^*(s,a)}{\sum_{a\in\mathbb{A}} y_i^*(s,a)}, & \text{if } \sum_{a\in\mathbb{A}} y_i^*(s,a) > 0, \\ \frac{1}{|\mathbb{A}|}, & \text{if } \sum_{a\in\mathbb{A}} y_i^*(s,a) = 0, \end{cases} \tag{7}$$

where $\bar{\pi}_i^*(a \mid s)$ is the probability of taking action $a$ given that the arm's current state is $s$. Note that due to heterogeneity, this optimal single-armed policy $\bar{\pi}_i^*$ can be different for different arms.

The rationale behind these policies is as follows. If each arm $i$ individually follows its optimal single-armed policy $\bar{\pi}_i^*$, then the average reward per arm (total reward divided by $N$) achieves the upper bound $R_N^{\text{rel}}$ given by the LP. However, this strategy only guarantees that the budget constraints are satisfied in a *time-average* sense, rather than conforming to the *hard* constraints in the original $N$-armed WCMDP. Thus, having each arm follow its optimal single-armed policy is not a valid policy for the original $N$-armed problem. Nevertheless, these optimal single-armed policies $\bar{\pi}_i^*$'s serve as a guide for how the arms should ideally behave to maximize rewards. The ID policy uses the $\bar{\pi}_i^*$'s as a reference. It is then designed to ensure that even under the hard budget constraints, most arms follow their optimal single-armed policies most of the time, yielding a diminishing gap to $R_N^{\text{rel}}$ in reward.

**ID reassignment.**   We first define a few quantities that will be used in the ID reassignment algorithm. For each arm $i \in [N]$ and each cost type $k \in [K]$, the expected cost under the optimal single-armed policy is defined as $C_{k,i}^* = \sum_{s\in\mathbb{S},a\in\mathbb{A}} y_i^*(s,a) c_{k,i}(s,a)$. Based on $C_{k,i}^*$'s, we divide the budget constraints into *active* constraints and *inactive* constraints as follows. For each cost type $k \in [K]$, we say the type-$k$ budget constraint is *active* if

$$\sum_{i\in[N]} C_{k,i}^* \geq \frac{\alpha_k}{2} N, \tag{8}$$

and *inactive* otherwise. Let $\mathcal{A} \subseteq [K]$ denote the set of cost types corresponding to active constraints. Note that replacing $\alpha_k N/2$ with any constant and strict fraction of $\alpha_k N$ will not change the results.

---

**Algorithm 2** ID policy with reassignment

---

1: **Input:** $N$-armed WCMDP instance $(\mathcal{M}_i)_{i\in[N]}$
2: **Preprocessing:**
3:    Solve the LP in (5) and obtain the optimal state-action frequencies $(y_i^*(s,a))_{i\in[N],s\in\mathbb{S},a\in\mathbb{A}}$
4:    Calculate the optimal single-armed policies $(\bar{\pi}_i^*)_{i\in[N]}$ using (7)
5:    Perform ID reassignment using Algorithm 1
6: **Real-time:**
7: **for** $t=0,1,2,\cdots$ **do**
8:    Sample ideal actions $\widehat{A}_{i,t} \sim \bar{\pi}_i^*(\cdot \mid S_{i,t})$ for all $i \in [N]$
9:    $I \leftarrow 1$
10:    **while** $\sum_{i\in[I]} c_{k,i}(S_{i,t}, \widehat{A}_{i,t}) \leq \alpha_k N, \forall k \in [K]$ **do**
11:       For arm $I$, take action $A_{I,t} = \widehat{A}_{I,t};\quad I \leftarrow I+1$
12:    For each arm $i \in \{I, I+1, \ldots, N\}$, take action $A_{i,t} = 0$

---

Based on the costs $C_{k,i}^*$'s and the active constraints, the ID reassignment algorithm rearranges arms so that the cost incurred by each contiguous segment of arms is approximately proportional to the length of the segment. We give a brief explanation of how the reassigned IDs affect the execution of the policy at the end of this section. The algorithm is formally described in Algorithm 1, with more details and properties provided in Appendix D of [54]. In the rest of the paper, we use the *reassigned IDs* to refer to arms, i.e., arm $i$ refers to the arm whose new ID assigned by Algorithm 1 is $i$.

**Constructing ID policy.**    We are now ready to describe our generalized ID policy, formally described in Algorithm 2. The policy begins with a one-time preprocessing phase: we solve the associated LP, construct the optimal single-armed policies, and reassign arm IDs using the ID reassignment algorithm (Algorithm 1). After the preprocessing, the policy proceeds at each time step $t$ as follows. For each arm $i$ (where $i$ is the reassigned ID), we first sample an action $\widehat{A}_{i,t}$, referred to as an *ideal action*, from the optimal single-armed policy $\bar{\pi}_i^*(\cdot \mid S_{i,t})$. We then attempt to execute these ideal actions, i.e., set the real actions equal to the ideal actions, in ascending order of arm IDs, starting from $i=1$, then $i=2$, and so on. We continue the attempt until we have used up at least one type of cost budget, at which point we let the remaining arms take action $0$ (the no-cost action).

This ID policy is a natural generalization of the ID policy designed for homogeneous restless bandits [27]. At a high level, using arm IDs to decide the priority order for executing ideal actions guarantees that a subset of arms (those with smaller IDs) can persistently follow their ideal actions. This persistency gives these arms time to converge to their optimal state-action frequencies, which in turn allows their instantaneous costs to converge to steady-state values. This convergence creates slack in the budget constraints, thereby allowing more arms to follow their ideal actions. In contrast, if we do not use IDs but instead randomly select a subset of arms to follow their ideal actions, the convergence may be disrupted. Indeed, [25] provides an example where this randomized strategy fails to achieve asymptotic optimality in restless bandits.

When there is no ambiguity, we refer to the ID policy with reassignment simply as the ID policy.

**How the reassigned IDs affect the policy execution.**    In the heterogeneous setting, arms differ in their cost consumptions under their optimal single-armed policies. The ID reassignment algorithm is designed to prevent "plateaus" in cumulative cost as we progress from smaller to larger IDs. If such a plateau exists, the subset of arms allowed to follow their ideal actions (determined by the budget constraints) can become sensitive to the randomness in action sampling, potentially leading to performance instability. The ID reassignment algorithm ensures a regularity property of the cost consumptions, stated in Lemma 2 in Appendix D of [54], which eliminates such plateaus.

## 4    Main results and technical overview

Before we present the main results, we first state our main assumption. This assumption is for the optimal single-armed policies $\bar{\pi}_i^*$'s. Note that each $\bar{\pi}_i^*$ is a stationary Markov policy. Therefore, under this policy, the state of arm $i$ forms a Markov chain. Let the transition probability matrix of this

Markov chain be denoted as $P_i = (P_i(s, s'))_{s \in \mathbb{S}, s' \in \mathbb{S}}$, where the row index is the current state $s$ and the column index is the next state $s'$. Then $P_i(s, s')$ can be written as

$$P_i(s, s') = \sum_{a \in \mathbb{A}} \mathbb{P}_i(s' \mid s, a)\bar{\pi}_i^*(a \mid s). \tag{9}$$

One can verify that the stationary distribution of this Markov chain is $\mu_i^* = (\mu_i^*(s))_{s \in \mathbb{S}}$ with $\mu_i^*(s) = \sum_{a \in \mathbb{A}} y_i^*(s, a)$, which we refer to as the *optimal state distribution* for arm $i$. Let $\tau_i$ be the *mixing time* of this Markov chain, defined as

$$\tau_i = \max_{s \in \mathbb{S}} \min \left\{ t \in \mathbb{N} \colon \left\| P_i^t(s, \cdot) - \mu_i^*(\cdot) \right\|_1 \leq 1/e \right\}, \tag{10}$$

where $P_i^t$ is the $t$-step transition probability matrix. The mixing time $\tau_i$ is finite if the Markov chain $P_i$ is unichain (one recurrent class, possibly with transient states) and aperiodic.

**Assumption 1.** *For each arm $i \in \mathbb{N}_+$, the induced Markov chain under the optimal single-armed policy $\bar{\pi}_i^*$ is an aperiodic unichain. Furthermore, the mixing times of these Markov chains have a uniform upper bound; i.e., there exists a positive $\tau$ such that for all $i \in \mathbb{N}_+$,*

$$\tau_i \leq \tau. \tag{11}$$

We remark that in the homogeneous or typed heterogeneous settings, once we make the aperiodic unichain assumption in Assumption 1, the uniform upper bound on mixing times automatically exists.

Next, we state our main theorem, Theorem 1, whose proof is provided in Appendix E of [54].

**Theorem 1.** *Consider an $N$-armed WCMDP problem satisfying Assumption 1, with initial system state $\boldsymbol{S}_0$. Let policy $\pi$ be the ID policy with reassignment (Algorithm 2). Then the optimality gap of $\pi$ is bounded as*

$$R^*(N, \boldsymbol{S}_0) - R(\pi, \boldsymbol{S}_0) \leq \frac{C_{\mathrm{ID}}}{\sqrt{N}},$$

*where $C_{\mathrm{ID}}$ is a positive constant independent of $N$.*

We re-emphasize that our proposed ID policy with reassignment is the first efficiently computable policy that achieves an $O(1/\sqrt{N})$ optimality gap for fully heterogeneous average-reward WCMDPs. In contrast, the best-known optimality gap for efficiently computable policies for average-reward WCMDPs is $o(1)$, achieved only under restrictive budget constraints and typed-heterogeneity.

We comment that the primary goal of this paper is to characterize the optimality gap in terms of its order in $N$, which is in line with the main focus of the large body of prior work on restless bandits and WCMDPs. While our analysis also gives an explicit expression for the constant $C_{\mathrm{ID}}$, which shows that $C_{\mathrm{ID}} = O(K^5 \max\{r_{\max}, c_{\max}\}^7 \tau^4 / \alpha_{\min}^6)$, we have not attempted to optimize its dependence on other problem parameters, either through refined analysis or alternative policy design.

*Remark* 1 (Generalization of result). Our result can be generalized to the setting where a $g(N)$ fraction of arms have unbounded mixing times, and the mixing times of the remaining arms scale with $N$ with an upper bound $\tau$. In this case, we can modify the ID policy by reassigning this $g(N)$ fraction of arms the largest IDs, effectively ignoring these arms. Applying Theorem 1 to the remaining arms then implies $R^*(N, \boldsymbol{S}_0) - R(\pi, \boldsymbol{S}_0) \leq O\big(K^5 \max\{r_{\max}, c_{\max}\}^7 \tau^4 / (\alpha_{\min}^6 \sqrt{N}) + r_{\max} g(N)\big)$. Consequently, this modified policy is asymptotically optimal when $g(N) = o(1)$ and $\tau = o(N^{1/8})$.

Adapting the proof of Theorem 1 also gives the following finite-time bound (see Appendix I of [54]).

**Proposition 1** (Finite-time bound). *Under the same conditions as Theorem 1, we have for any $T \geq 1$,*

$$R^*(N, \boldsymbol{S}_0) - \frac{1}{TN} \sum_{t=0}^{T-1} \sum_{i \in [N]} \mathbb{E}\left[ r_i(S_{i,t}^\pi, A_{i,t}^\pi) \right] \leq \frac{C_{\mathrm{ID}}}{\sqrt{N}} + \frac{C_{\mathrm{finite}}}{T}, \tag{12}$$

*where $C_{\mathrm{finite}}$ is another positive constant independent of $N$.*

**Technical overview**

Our technical approach uses the Lyapunov drift method, which has found widespread applications in queueing systems, Markov decision processes, reinforcement learning, and so on. While the basic

framework of the drift method is standard, the *key challenge* lies in constructing the right Lyapunov function with the desired properties, where the difficulty is exacerbated by the full heterogeneity of the problem under study. Our construction of such a Lyapunov function is highly novel, yet still natural. We reiterate that fully heterogeneous, high-dimensional stochastic systems are poorly understood in the existing literature. Our approach opens up the possibility of analyzing the steady-state behavior of such systems through the Lyapunov drift method.

In the remainder of this section, we consider the ID policy, also referred to as the policy $\pi$. Let $\boldsymbol{X}_t$ denote the system state under it, with the superscript $\pi$ omitted for brevity. To make this overview more intuitive, here let us assume that $\boldsymbol{X}_t$ converges to its steady state $\boldsymbol{X}_\infty$ in a proper sense such that taking expectations in steady state is the same as taking time averages. However, note that our formal results do not need this assumption and directly work with time averages. We call a function $V$ a Lyapunov/potential function if it maps each possible system state to a nonnegative real number.

**General framework of the drift method.** Here we briefly describe the general framework of the drift method when applied to our problem. The goal is to construct a Lyapunov function $V$ such that

(**C1**) $R_N^{\text{rel}} - R(\pi, \boldsymbol{S}_0) \leq C_1 \mathbb{E}\left[V(\boldsymbol{X}_\infty)\right]/N + O(1/\sqrt{N})$ for some constant $C_1$;

(**C2**) (Drift condition) $\mathbb{E}\left[V(\boldsymbol{X}_{t+1}) \mid \boldsymbol{X}_t\right] - V(\boldsymbol{X}_t) \leq -C_2 V(\boldsymbol{X}_t) + O(\sqrt{N})$ for a constant $C_2$.

The drift condition requires that on average, the value of $V$ approximately decreases (ignoring the additive $O(\sqrt{N})$) after a time step. The drift condition implies a bound on $\mathbb{E}\left[V(\boldsymbol{X}_\infty)\right]$. To see this, let $\boldsymbol{X}_t$ follow the steady-state distribution, which means $\boldsymbol{X}_{t+1}$ also follows the steady-state distribution, and take expectations on both sides of the inequality. Then we get $0 = \mathbb{E}\left[V(\boldsymbol{X}_{t+1})\right] - \mathbb{E}\left[V(\boldsymbol{X}_t)\right] \leq -C_2 \mathbb{E}\left[V(\boldsymbol{X}_t)\right] + O(\sqrt{N})$, which implies $\mathbb{E}\left[V(\boldsymbol{X}_\infty)\right] = \mathbb{E}\left[V(\boldsymbol{X}_t)\right] = O(\sqrt{N})$. Combining this with (**C1**) proves the desired $O(1/\sqrt{N})$ upper bound on the optimality gap.

**Key challenge: constructing Lyapunov function.** We highlight this challenge by contrasting the homogeneous setting and the heterogeneous setting. In the *homogeneous* setting, there is only one optimal state distribution, $\mu^*$. The Lyapunov function in [27] is defined based on the distance between the *empirical state distribution* across arms and $\mu^*$. Specifically, it is based on a set of functions $(h(\boldsymbol{X}_t, D))_{D \subseteq [N]}$ defined as:

$$h(\boldsymbol{X}_t, D) = \|\boldsymbol{X}_t(D) - m(D)\mu^*\|, \tag{13}$$

where $\boldsymbol{X}_t(D) = (X_t(D, s))_{s \in \mathbb{S}}$ denotes within $D$, the number of arms in each state $s$, divided by $N$; $m(D) = |D|/N$; and the norm $\|\cdot\|$ is a properly defined norm. The idea is that if all arms in $D$ follow the optimal single-armed policy, the state distribution of each arm in $D$ gets closer to $\mu^*$, and thus $\boldsymbol{X}_t(D)$ gets closer to $m(D)\mu^*$ over time.

In the *heterogeneous* setting, we also want to construct a Lyapunov function $h(\boldsymbol{X}_t, D)$ to witness the convergence of any set of arms $D$ if they follow the optimal single-armed policies. However, unlike the homogeneous setting, now it no longer makes sense to aggregate arm states into an empirical state distribution, since each arm's dynamics is distinct. Instead, our Lyapunov function considers $X_{i,t} - \mu_i^*$, where recall $X_{i,t}(s)$ is the indicator that arm $i$'s state is $s$ at time $t$. A naive first attempt is to construct the Lyapunov function from the pointwise distances, $\|X_{i,t} - \mu_i^*\|$ for each arm $i$, with a properly defined norm $\|\cdot\|$. However, the pointwise distances are very noisy: $\|X_{i,t} - \mu_i^*\|$ could be large even when the state of arm $i$ independently follows the distribution $\mu_i^*$ for each $i$, a situation when we should view the set of arms as already converged.

Intuitively, to make the Lyapunov function properly reflect the convergence of the set of arms (referred to as "the system" in the rest of the section) following the optimal single-armed policies, we would like it to depend less strongly on the state of each individual arm and focus more on the collective properties of the whole system. Our idea is to *project* the system state onto a properly selected set of *feature vectors*, and construct the Lyapunov function based on how far these projections are from the projections of the optimal state distributions $(\mu_i^*)_{i \in [N]}$. Then what features of the system state do we need to determine whether it has converged in a proper sense? The first feature we consider is the instantaneous reward of the system, $\sum_{i \in D} \langle X_{i,t}, r_i^* \rangle$, where $r_i^* \in \mathbb{R}^\mathbb{S}$ is the reward function of arm $i$ under $\bar{\pi}_i^*$, and recall that the inner product is defined between two vectors whose entries correspond to states in $\mathbb{S}$. We also want to keep track of the $\ell$-step ahead expected reward, $\sum_{i \in D} \langle X_{i,t} P_i^\ell, r_i^* \rangle$,

for each $\ell \in \mathbb{N}_+$. Intuitively, if $\sum_{i \in D} \langle (X_{i,t} - \mu_i^*) P_i^\ell, r_i^* \rangle$ is small for each $\ell \in \mathbb{N}$, the reward of the system should remain close to that under the optimal state distributions $(\mu_i^*)_{i \in [N]}$ for a long time; conversely, if the state of each arm $i$ independently follows $\mu_i^*$, each of these features should be small as well. We also consider the $\ell$-*step ahead expected type-k cost* for each $\ell \in \mathbb{N}$ and $k \in [K]$ as features, defined analogously.

Combining the above ideas, for any set of arms $D$, we let the Lyapunov function $h(\boldsymbol{X}_t, D)$ be the supremum of the differences between $\boldsymbol{X}_t$ and $\mu^*$ in all the features directions defined above, under proper weightings:

$$h(\boldsymbol{X}_t, D) = \max_{g \in \mathcal{G}} \sup_{\ell \in \mathbb{N}} \left| \sum_{i \in D} \langle (X_{i,t} - \mu_i^*) P_i^\ell / \gamma^\ell, g_i \rangle \right|, \tag{14}$$

where $\gamma = \exp(-1/(2\tau))$ for $\tau$ defined in Assumption 1; each element $g \in \mathcal{G}$ is either $g = (r_i^*)_{i \in [N]}$, or corresponds to the type-$k$ cost for some $k \in [K]$ (See Appendix E of [54] for the definition of $\mathcal{G}$). Note that *dividing* each term by powers of $\gamma$ is another interesting trick, which induces a negative drift in $h(\boldsymbol{X}_t, D)$ under the optimal single-armed policies (See the proof of Lemma 3 in [54]).

Now with the set of functions $(h(\boldsymbol{X}_t, D))_{D \subseteq [N]}$ defined, we generalize the idea of focus sets in [27] to convert $(h(\boldsymbol{X}_t, D))_{D \subseteq [N]}$ into a Lyapunov function $V(\boldsymbol{X}_t)$. We prove that $V$ satisfies (C1) and (C2) using the structure of $(h(\boldsymbol{X}_t, D))_{D \subseteq [N]}$.

*Remark* 2. The idea for constructing $h(\boldsymbol{X}_t, D)$ is potentially useful for analyzing other heterogeneous stochastic systems. At a high level, projecting the system state onto a set of feature vectors (and their future expectations) can be roughly viewed as aggregating system states whose relevant performance metrics *remain close for a sufficiently long time*. This idea provides a new way to measure the distance between two system states in a heterogeneous system, and this distance notation enjoys similar properties as that in a homogeneous system, without resorting to symmetry.

## 5 Experiments

In this section, we perform two sets of experiments to illustrate the numerical performance of the proposed ID policy for fully heterogeneous WCMDPs.

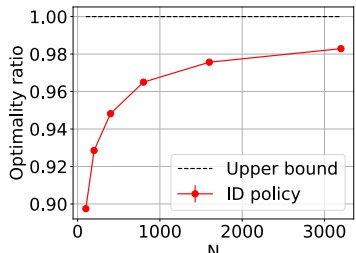

In the first set of experiments, we demonstrate the asymptotic optimality of the ID policy. We increase the number of arms as $N \in \{100, 200, 400, 800, 1600, 3200\}$. Each arm's MDP has 10 states and 4 actions, with parameters generated uniformly at random in a proper sense. The $N$-armed problem has 4 budget constraints, with cost functions also generated randomly. More details are provided in Appendix B.1. We simulate the policy for $2 \times 10^4$ time steps over 4 replications for each $N$. To illustrate the performance more clearly, we measure the

Figure 1: Asymptotic optimality of ID policy.

*optimality ratio*, defined as the ratio between the long-run average reward achieved by a policy and the LP relaxation upper bound $R_N^{\text{rel}}$. Confidence intervals are calculated using the batch means method with a batch size of 4000, but they are typically too small to be visible on figures. Figure 1 shows that the optimality ratio of the ID policy becomes increasingly close to 1 as $N$ increases.

In the second set of experiments, we compare the ID policy with the ERC policy proposed in [46]. As discussed in Section 1, only a few prior papers address heterogeneous WCMDPs. Among them, [46] considers the most general setting, but is still limited to a single-budget constraint, state-independent costs, and typed heterogeneity. To make a fair comparison, we evaluate both policies under this special case, while keeping all other settings the same as in the first set of experiments. ID policy turns out to have a slight improvement over ERC policy in some instances, one of which is shown in Figure 2, and has comparable performance in others. Importantly, unlike ERC, the ID policy applies to far more general classes of WCMDPs.

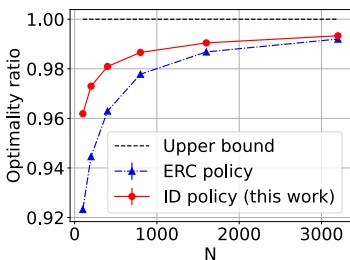

Figure 2: ID policy vs ERC policy.

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

# A   Detailed review on related work

In this section, we provide a more detailed, though still non-exhaustive, review of the literature. We mainly focus on theoretical work with formal performance guarantees, leaving out the extensive body of work with empirical results. We begin by surveying papers with the same reward criterion as ours, i.e., infinite-horizon average-reward criterion. In this setting, we first review existing work on restless bandits (RBs), which is an extensively studied special case of WCMDPs. We then give a more detailed review of existing results on WCMDPs. Next, we turn to work that considers other reward criteria: the finite-horizon total-reward criterion and the infinite-horizon discounted-reward criterion. Finally, we briefly mention other problems that are related to WCMDPs.

**Infinite-horizon average-reward RBs.**   For the homogeneous arm setting, the first asymptotic optimality result for average-reward homogeneous RBs is established by Weber and Weiss [42]: they show that the *Whittle index policy* [44] achieves an $o(1)$ optimality gap as the number of arms $N$ goes to infinity. There are three key assumptions in [42]: indexability, the global attractor property, and the aperiodic-unichain condition. These assumptions have been gradually relaxed in the subsequent papers. In particular, [39] proposes a class of priority policies based on an LP relaxation. This class of policies, later referred to as the *LP-Priority policies*, generalizes the Whittle index policy. Each LP-Priority policy achieves an $o(1)$ optimality gap without requiring indexability. The work [25] is the first one that breaks the global attractor property assumption. The authors propose a policy named Follow-the-Virtual-Advice (FTVA), which achieves an $O(1/\sqrt{N})$ optimality gap under an assumption named the Synchronization Assumption; there exist problem instances that satisfy the Synchronization Assumption but do not satisfy the global attractor property. Later work [27] further relaxes the conditions and only requires the aperiodic-unichain condition to achieve an $O(1/\sqrt{N})$ optimality gap. More recently, Yan [47] proposes the *align-and-steer policy*, which further weakens the aperiodic-unichain condition and achieves an $o(1)$ optimality gap.

Parallel to relaxing the assumptions for asymptotic optimality, another line of work has focused on improving the optimality gap beyond $O(1/\sqrt{N})$ under slightly stronger assumptions [14, 15, 26]. Specifically, Gast et al. [14] show that the Whittle index policy has an $O(\exp(-cN))$ optimality gap for some constant $c > 0$. In addition to indexability and the aperiodic-unichain condition, [14] also requires a stronger version of the global attractor property named Uniform Global Attractor Property (UGAP), and a condition called non-singularity. Subsequently, Gast et al. [15] show that LP-Priority policies achieve $O(\exp(-cN))$ optimality gaps assuming the aperiodic-unichain condition, UGAP, and a non-degenerate condition that is equivalent to non-singularity. More recently, Hong et al. [26] propose a *two-set policy* that also achieves an $O(\exp(-cN))$ optimality gap while replacing UGAP of [15] with a much weaker condition named local stability.

Among the aforementioned work, [39] and [25] have addressed the heterogeneous arm setting. The setting studied in [39] is the typed heterogeneous setting, where the $N$ arms are divided into a constant number of types as $N \to \infty$. The paper [25] includes an extension to the fully heterogeneous setting. In particular, the proposed FTVA policy generalizes to the fully heterogeneous setting and leads to an optimality gap of $O(\overline{\tau}_{\max}^{\text{sync}}/\sqrt{N})$, where $\overline{\tau}_{\max}^{\text{sync}}$ is the maximum of a quantity called the synchronization time across all arms. Therefore, for this result to yield asymptotic optimality, there needs to be a further assumption that $\overline{\tau}_{\max}^{\text{sync}} = o(\sqrt{N})$. We re-emphasize that the FTVA policy does not generalize to WCMDPs. The main reason is that FTVA heavily relies on the fact that an RB only constrains the number of pulls, while a WCMDP has budget constraints on cost functions each depending on both the state and action of an arm.

**Infinite-horizon average-reward WCMDPs.**   Work on average-reward WCMDPs remains relatively scarce, and to our knowledge, fully heterogeneous WCMDPs have yet to be addressed. Compared to RBs, a WCMDP allows multiple actions for each arm and multiple cost constraints, where each cost function is a function of both the state and the action of an arm. The line of research [22, 24, 39, 46] has generalized the action space and cost model of RBs to WCMDPs, and some of them allow for typed heterogeneity. In particular, Hodge and Glazebrook [24] generalize the Whittle index policy to homogeneous WCMDPs with a single constraint and multiple actions, where each action represents a different activation level and has a different cost. Verloop [39] extends the LP-Priority policies to typed heterogeneous WCMDPs with a single constraint and multiple actions, but requires each action to have the same cost. Then Xiong et al. [46] propose another index policy

for typed heterogeneous WCMDPs with a single constraint, and allow each action to have a different cost. In the three papers above [24, 39, 46], $o(1)$ optimality gaps have been proved under a similar set of assumptions as in most restless bandit papers, i.e., aperiodic-unichain (or irreducibility) condition, global attractor property, and a generalized indexability condition if the policy is Whittle index. Finally, Goldsztajn and Avrachenkov [22] consider homogeneous WCMDPs with multiple actions and multiple cost constraints with general cost functions, and propose a class of policies with $o(1)$ optimality gaps under a weaker-than-standard aperiodic-unichain condition.

**Finite-horizon total-reward RBs and WCMDPs.** Next, we review the asymptotic optimality results for finite-horizon total-reward RBs and WCMDPs. The finite-horizon setting is better understood than the average-reward setting, partly because the analysis in the finite horizon is not hindered by the technical conditions arising in average-reward MDPs, such as the unichain condition and the global attractor property. On the other hand, computing asymptotically optimal policies for the finite-horizon setting is more complicated, requiring a careful optimization of the transient sample paths.

Hu and Frazier [28] propose the first asymptotically optimal policy for finite-horizon homogeneous RBs, which achieves an $o(1)$ optimality gap without any assumptions.[2] Since then, researchers have established asymptotic optimality in more general settings [8, 10, 12, 17, 49]. Among these papers, the most general setting is addressed by Brown and Zhang [10], where the authors consider fully heterogeneous WCMDPs; they obtain $O(1/\sqrt{N})$ optimality gaps for a naive fluid policy and a reoptimization-based fluid policy, among a few other results to be reviewed in the next paragraphs. Notably, there is also a further generalization of fully heterogeneous WCMDPs, which involves an exogenous state that affects all arms' transitions, rewards, and constraints; Brown and Zhang [9] propose this setting, where they achieve an $O(1/\sqrt{N})$ optimality gap using a dynamic fluid policy.

Another line of work has improved the optimality gap beyond the order $O(1/\sqrt{N})$ by making an additional assumption called non-degeneracy. Specifically, Zhang and Frazier [51] establish an $O(1/N)$ optimality gap in non-degenerate homogeneous RBs. Gast et al. [15] then propose a different policy for the same setting that improves the optimality gap to $O(\exp(-cN))$. Later, Gast et al. [16] and Brown and Zhang [10] establish $O(1/N)$ optimality gaps for homogeneous and typed heterogeneous WCMDPs, respectively, assuming non-degeneracy. More recently, Zhang [50] proposes a policy for fully heterogeneous WCMDPs; the optimality gap bound of the policy interpolates between $O(1/\sqrt{N})$ and $O(1/N)$ as the degree of non-degeneracy varies, unifying the performance bounds in the degenerate and non-degenerate cases.

Despite the generality of the settings and the fast diminishing rate of the optimality gaps as $N \to \infty$, most of the optimality gaps in the finite-horizon setting depend *super-linearly* on the time horizon, so they do not carry over to the infinite-horizon average-reward setting. There are two exceptions, [10, 16], which achieve optimality gaps that depend linearly on the time horizon under some special conditions: [10] requires all entries of the transition kernels to be bounded away from zero; [16] assumes an ergodicity property, which requires two arms in any different states to synchronize in a fixed number of steps under *any sequence of actions* with a positive probability. However, without these conditions, the optimality gaps in [10, 16] depend quadratically on the time horizon. Apart from having distinct optimality gap bounds, all existing algorithms in the finite-horizon setting need to (sometimes repeatedly) solve LPs whose number of variables scales with the time horizon, so they cannot be directly adapted to the infinite-horizon average-reward setting.

**Infinite-horizon discounted-reward RBs and WCMDPs.** Asymptotic optimality has also been established for RBs and WCMDPs under the infinite-horizon discounted-reward criterion. In particular, Brown and Smith [8] establish an $O(N^{\log_2(\sqrt{\gamma})})$ optimality gap for fully heterogeneous WCMDPs when $\gamma \in (1/2, 1)$. Subsequently, Ghosh et al. [17], Zhang and Frazier [52] obtain $O(1/\sqrt{N})$ optimality gaps for homogeneous and typed heterogeneous RBs, and Brown and Zhang [10] establish the same order of optimality gap for fully heterogeneous WCMDPs. Similar to the finite-horizon setting, most of these optimality gaps depend super-linearly on the effective time horizon $1/(1 - \gamma)$

---

[2]Here, we measure the optimality gap in terms of the reward per arm, to be consistent with our convention. However, in the papers on the finite-horizon total-reward setting, it is also common to measure the optimality gap in terms of the total reward of all arms, which differs from ours by a factor of $N$. We also adopt the same convention when reviewing the papers on the infinite-horizon discounted-reward setting.

unless special conditions hold [10], so they do not carry over to the infinite-horizon average-reward setting. The policies here also require solving LPs whose complexities scale with the effective time horizon.

**Restful bandits, stochastic multi-armed bandits.** A special case of RB is the restful bandit (also referred to as nonrestless bandits, rested bandits, or Markovian bandits), where an arm's state does not change if it is not pulled. The restful bandit problem has been widely studied, where the celebrated Gittins index policy is proven to be optimal [4, 19, 20, 37, 38, 41, 43]. We refer the readers to [18] for a comprehensive review of Gittins index and restful bandits. Another related topic is the stochastic multi-armed bandit (MAB) problem, which has been extensively studied; see the book [31] for a comprehensive overview. The key distinction between MABs and RBs is that arms are stateless in MABs, but stateful in RBs. Consequently, MAB becomes trivial with known model parameters, whereas RB is still non-trivial.

# B Experimental details

In this appendix, we provide details of the two WCMDP instances considered in Section 5, and the definition of the baseline policy, the ERC policy from [46]. The complete code for these experiments is available on GitHub [53], and all results can be reproduced within 24 hours on a standard PC (e.g., 6-Core Intel Core i7).

## B.1 WCMDP instance generation

**Details of the WCMDP instance 1 (In Figure 1).** $|\mathbb{S}| = 10$, $|\mathbb{A}| = 4$, $K = 4$. For each $i \in [N]$, $s \in \mathbb{S}$, and $a \in \mathbb{A}$, $r_i(s, 0) = 0$, and $r_i(s, a)$ is independently sampled from the uniform distribution over $[0, 1]$ for each $a \neq 0$; $P_i(s, a, \cdot)$ is independently sampled from the uniform distribution over the probability simplex. As for the cost function, for each $i \in [N]$, $s \in \mathbb{S}$, $a \in \mathbb{A}$, and $k \in [K]$, we have $c_{k,i}(s, 0) = 0$, and $c_{k,i}(s, a)$ is independently sampled from the uniform distribution over $[0, 1]$ for each $a \neq 0$. For each $k \in [K]$, $\alpha_k$ is uniformly sampled from $\{0.05, 0.1, 0.15, \ldots, 0.45\}$, i.e., the uniform distribution over integer multiples of 0.05 in the interval $(0, 0.5)$.

**Details of the WCMDP instance 2 (In Figure 2).** This instance is typed heterogeneous with 10 types, with equal fraction of arms in each type. Each arm has $|\mathbb{S}| = 10$ states, $|\mathbb{A}| = 4$ actions. For each type, the reward function and transition kernel are generated in the same way as each arm in ithe first instance. The budget is also sampled from the same distribution as the first instance. There is a single budget constraint. The cost function depends only on the action, consistent with the setting in [46]; the cost of each of action $a \in \mathbb{A}$ is sampled from the uniform distribution over $[0, 1]$.

## B.2 Definition of ERC policy

The ERC policy [46] solves the same LP in (5) to define the single-armed policies $\bar{\pi}_i^*(a|s)$ as in (7). At each time step, the policy computes an index for each arm-state pair $(i, s) \in [N] \times \mathbb{S}$:

$$\mathcal{I}(i, s) = \sum_{a \in \mathbb{A}} \bar{\pi}_i^*(a|s) r_i(s, a).$$

It then iterates through the arms in descending order of these indices. For each arm, it samples an action $a \sim \bar{\pi}_i^*(\cdot \mid s)$ and applies it only if the budget is sufficient. If not, it defaults to action 0.

