# OpenReview forum: "Projection-based Lyapunov method for fully heterogeneous weakly-coupled MDPs"
_NeurIPS.cc/2025/Conference — NeurIPS 2025 spotlight_

### Official Review · Reviewer_9BWY · 2025-06-03

**Clarity:** 4
**Significance:** 3
**Originality:** 3
**Rating:** 5
**Confidence:** 3

**Summary:**

This paper analyzes weakly-coupled Markov decision processes (WCMDPs), a classical framework for solving large decision-making problems, under the mean-payoff criteria. An algorithm is given with suboptimality guarantees of $O(1/\sqrt{N})$, where $N$ is the number of components of the weakly-coupled model.

**Questions:**

**Questions**

- What is the relation between WCMDPs and factored MDP or constrained MDP?
- How to go beyond unichain? Did the authors consider weakly-communicating models?
- l310: “let us assume …”: isn’t this implied by Assumption 1 ?
- l363, quote: “Note that dividing each term by powers of $\phi$ is another interesting trick, which makes the Lyapunov function strictly contractive under the optimal single-armed policies, as demonstrated in the proof of Lemma 3.” What do you mean by “strictly contractive”? Do you just mean contractive (Lipschitz-continuity with constant strictly smaller than 1)? Also in the proof of Lemma 3 (app E.2) I only found the proof of the Lipschitzness, not of the contraction property. Please clarify.

**Suggestions**

- Maintaining feasibility in the hard-constraint sense is crucial. The fact that the authors manage to do it is pretty strong and perhaps should be emphasized earlier in the paper / in the exposition of the results; I was wondering about this until the middle of page 6 (paragraph line 225 - 232).
- What does “ID” mean (line 49 and onward)? Perhaps introduce the full term the first time.
- Unichain assumption should be stated in the abstract and intro. It is crucial, but it only appears on the last-but-one page (page 8).

**Ethical Concerns:**

["NO or VERY MINOR ethics concerns only"]

**Final Justification:**

I want to thank the reviewers for taking the time to address my comments. My assessment of the paper remains very positive, and I will keep my score. I encourage the authors to enforce the minor (in my opinion) stylistic improvements that I suggested.

**Limitations:**

Yes.

**Paper Formatting Concerns:**

No.

**Quality:**

4

**Strengths And Weaknesses:**

**Strengths**

- The paper is very well-written, much better than most submissions I've reviewed this year.
- The results are strong and the technics are new. This is a good paper.

**Weaknesses** I don't see significant weaknesses. I only have a few questions/comments that could improve the presentation of the results and the positioning of the paper with respect to the rest of the literature on MDPs.

---

> ### Author Rebuttal · Authors · 2025-07-30
>
> > Question 1: What is the relation between WCMDPs and factored MDP or constrained MDP?
> >
>
> Thanks for raising this interesting question. For the factored MDP and WCMDPs, while they both consist of some “loosely-coupled” components, their components are coupled in different ways: In the factored MDP, the components lie in a network, and the state transitions of nearby components are correlated. In contrast, the arms in WCMDPs have *independent* state transitions conditioned on the previous states and actions, but the joint actions of the arms are subject to a global constraint. It is unclear to us whether one problem class contains the other, or if there are any formal connections between these two classes.
>
> WCMDPs is also different from the constrainted MDP: in the WCMDPs, the instantaneous costs in every time step are subject to constraints, whereas in the constrained MDP, the long-run average costs are subject to constraints. Nevertheless, the LP relaxation of the WCMDPs is equivalent to a constrained MDP, with (5b) being the long-run average constraints on each type of cost.
>
> > Question 2: How to go beyond unichain? Did the authors consider weakly-communicating models?
> >
>
> Going from unichain to weakly-communicating models (or more general) is non-trivial; even for the homogeneous case, asymptotically optimal policies are found only very recently by [22, 47]. On a high-level, their policies maintain the distribution of a subset of arms whose empirical state distribution is close to the optimal stationary distribution $\mu^*$, while trying to control the rest of the arms so that they can be gradually merged into the first subset. This idea might be useful for the fully heterogeneous setting, although an obvious obstacle is to remove the dependency on the empirical state distribution, and it will be an interesting future work to see if our projection idea can work here.
>
> > Question 3: l310: “let us assume …”: isn’t this implied by Assumption 1 ?
> >
>
> Thanks for bringing up this point. We would like to clarify that Assumption 1 only implies that each single-armed MDP mix to the steady state under the policy $\bar{\pi}_i^*$, whereas on Line 310 we assume that the joint state of all arms has a limiting distribution.
>
> > Question 4: l363, What do you mean by “strictly contractive”? Do you just mean contractive (Lipschitz-continuity with constant strictly smaller than 1)? Also in the proof of Lemma 3 (app E.2) I only found the proof of the Lipschitzness, not of the contraction property. Please clarify.
> >
>
> Thanks for pointing out this confusion. By “strictly contractive”, we are referring to the drift condition (C2) on page 8, i.e., $h(\mathbf{X}\_t P) \leq (1-C\_2) h(\mathbf{X}\_t) + O(\sqrt{N})$, and it is formally stated in the second bullet point of Lemma 3. We used the term “contractive” because $1-C\_2 < 1$, but we will change the wording to avoid this confusion.
>
> > Suggestion 2: What does “ID” mean (line 49 and onward)?
> >
>
> Here ID is a short hand for identity, which are simply the integer numbers 1, 2, 3, … N assigned to the arms. We will clarify this name in the revision.
>
> We thank the reviewer for all the constructive suggestions.

---

> > ### Comment · Reviewer_9BWY · 2025-08-01
> > **Response to the authors**
> >
> > I want to thank the reviewers for taking the time to address my comments. My assessment of the paper remains very positive, and I will keep my score. I encourage the authors to enforce the minor (in my opinion) stylistic improvements that I suggested.

---

### Official Review · Reviewer_oixq · 2025-06-26

**Clarity:** 4
**Significance:** 2
**Originality:** 2
**Rating:** 5
**Confidence:** 3

**Summary:**

This paper considers the setting of weakly coupled MDPs (WCMDPs), which consist of $N$ arms, each representing a distinct individual MDP. The authors focus on the fully heterogeneous case, where each underlying MDP is governed by a different model. Considering all the models of the individual arms to be fully known, the objective is to maximize the overall reward while satisfying global constraints related to $K$ different cost functions. To this end, the authors propose a policy named "ID policy with reassignment", which operates in two phases. The first is the pre-processing phase: for each arm, compute the optimal single-arm policy that prescribes the ideal action the agent should take in each individual MDP. The second is the real-time phase: each MDP is assigned actions from its ideal policy as much as possible without violating the cost constraints, otherwise a neutral policy is applied. By leveraging a Lyapunov function specifically designed for this setting, the authors prove that the proposed policy achieves an asymptotic sub-optimality gap of order $\mathcal{O}(1/\sqrt{N})$, where $N$ is the number of arms.

**Questions:**

1. Do you have any insights on how results extend to the case in which we consider continuous sub-MDPs?
2. What are the main technical challenges in considering learning policies in unknown sub-MDPs? For instance, is would be possible to apply concepts by constrained reinforcement learning?
3. In the paragraph at lines 147--158, is it really necessary to indicate the policy $\pi$ when considering the vectors of MDPs states and actions?
4. Concerning the definition of $R^*(N,\boldsymbol{S}_{0})$, is it the optimal reward only or is it accounting for constraint satisfaction too?
5. In Lemma 1, what happens to constraint violation when employing the solution of the relaxed problem?

**Ethical Concerns:**

["NO or VERY MINOR ethics concerns only"]

**Final Justification:**

I believe this paper provides valuable contributions to the field.
After the rebuttal, all my concerns and doubts on some aspects of the work were resolved, for instance the ones regarding extension to learning and continuous settings, the ones related to the concept of optimality, and the ones related to the motivations for employing a tabular planning scenario. Finally, the authors conducted a toy simulation, which is in line with the theoretical nature of the proposed work, that I think to enforce the presented results.

**Limitations:**

The authors discuss widely the limitations of their work throughout the whole manuscript.

**Paper Formatting Concerns:**

The paper formatting respects the NeurIPS guidelines.

**Quality:**

4

**Strengths And Weaknesses:**

**Strengths**
The main strength of the paper lies in the quality of the manuscript. The paper is very well written; the authors take the time to discuss every introduced concept, the manuscript is self-contained, the related work is thoroughly reviewed, the notation is clear, and every symbol used is formally defined. Moreover, the authors consider a setting that reflects real-world scenarios, and the proposed ideas are intuitive and well presented. The paper appears to be theoretically sound, and the methodological analysis seems novel (at least to the best of my knowledge, although this is not precisely my area of expertise).

**Weaknesses**
The main weakness I found in the paper is that the authors assume full knowledge of all the tabular MDPs. This assumption makes the problem significantly easier, as it allows the proposed policy to solve each MDP exactly. Additionally, I believe the setting would benefit (and become more challenging) by introducing a finite budget for acting in the MDPs. The paper would also gain from a more detailed discussion of the motivational examples, which are just listed; elaborating on them could help the reader better appreciate the relevance of the proposed setting. That being said, I do not believe this is strictly a reinforcement learning paper, as it does not involve learning a policy through interaction with the environment(s). Furthermore, I would have appreciated at least a toy simulation, to provide a practical illustration of the theoretical results.

---

> ### Author Rebuttal · Authors · 2025-07-30
>
> > Weakness: The main weakness I found in the paper is that the authors assume full knowledge of all the tabular MDPs. This assumption makes the problem significantly easier, as it allows the proposed policy to solve each MDP exactly ...  I do not believe this is strictly a reinforcement learning paper, as it does not involve learning a policy through interaction with the environment(s).
> >
>
> We thank the reviewer for bringing up this point, as we also found the learning aspect of this problem interesting.  That said, we would like to argue that the planning setting itself with known parameters is an interesting setting for the two reasons below, and it needs to be better understood before we move onto the learning setting.
>
> First, the planning setting of WCMDPs and restless bandits has broad applications, especially in operations research, and there has been strong interest in developing better planning algorithms for more general settings.  We’d like to refer to the following survey for a comprehensive overview, and the references in our paper for this continued effort.
>
>     José Niño-Mora. Markovian restless bandits and index policies: A review. *Mathematics*, 11(7), 2023.
>
> Furthermore, planning algorithms can play an instrumental role for learning a policy.  For WCMDPs, if it is treated as a generic tabular MDP, then the sample complexity will be at least the state space size, which is exponential in N.  A more tractable approach could be estimate model parameters first, and then apply a good planning algorithm with the estimated parameters.  Such a model-based approach has been studied before for other RL problems.  This approach is promising for WCMDPs with the ID policy and our results in this paper, but it is non-trivial to establish rigorous guarantees and sample complexity bounds.  We will discuss the challenges in our response to Question 2.
>
> > Weakness (continued): Furthermore, I would have appreciated at least a toy simulation, to provide a practical illustration of the theoretical results.
> >
> >
> >
>
> We have performed simulation experiments on some random WCMDP instances. Below we display the simulation results on two representative instances. Each WCMDP instance is fully heterogeneous, whose arms are MDPs with independently generated parameters. For each arm, each row of the transition kernel is uniformly sampled from the probability simplex, each entry of the reward function and cost function is uniformly sampled from $[0, 1]$, and each budget parameter $\alpha_k$ is also uniformly sampled from $[0,1]$.
>
> The first instance has a single cost constraint, with $|\mathbb{S}| = 10$, $|\mathbb{A}| = 4$,. We compare the performance of our policy with the performance of a policy name ERC (“Expected Reward and pull arms constrained by activation Cost” ) [46]. The ERC policy is designed for the typed heterogeneous WCMDPs with a single cost constraint, but we can apply it to fully heterogeneous WCMDPs by regarding each arm as having a different type. We evaluate the “optimality ratio” of the policies, which is defined as the ratio between the long-run average reward $R(\pi, \mathbf{S}_0)$ and the optimal value of the LP relaxation $R^{\text{rel}}_N$.
>
> We simulate each of the two policies for $10^4$ time steps and $4$ replications for each $N\in \\{100,200,...1000\\}$. The result is displayed in the next table, where the confidence interval is calculated using the batch means method. We can see that the performance of these two policies are comparable, with the ID policy being slightly better.
>
> | N | 100 | 200 | 300 | 400 | 500 | 600 | 700 | 800 | 900 | 1000 |
> | --- | --- | --- | --- | --- | --- | --- | --- | --- | --- | --- |
> | ID policy (our work) | 0.9600 | 0.9732 | 0.9788 | 0.9814 | 0.9839 | 0.9855 | 0.9865 | 0.9874 | 0.9881 | 0.9887 |
> | Confidence interval width | 0.0011 | 0.0006 | 0.0005 | 0.0005 | 0.0003 | 0.0004 | 0.0003 | 0.0002 | 0.0003 | 0.0003 |
> | ERC policy | 0.9470 | 0.9643 | 0.9716 | 0.9760 | 0.9787 | 0.9808 | 0.9826 | 0.9836 | 0.9844 | 0.9851 |
> | Confidence interval width | 0.0013 | 0.0006 | 0.0003 | 0.0003 | 0.0003 | 0.0003 | 0.0003 | 0.0004 | 0.0003 | 0.0002 |
>
> The second instance has three cost constraints, with $|\mathbb{S}| = 10$, $|\mathbb{A}| = 2$,. We are not aware of any existing policies for the average-reward heterogeneous WCMDPs with multiple constraints, so we only evaluate the performance of our policy here.  We simulate $10^4$ time steps and $4$ replications for each $N\in \\{100, 200, 300, ... 1000, 2000, 3000, ... 5000\\}$. Here, we consider a larger range of $N$ to more clearly demonstrate the asymptotic optimality of our policy. As we can see from the following table, as the number of arms $N$ gets large, the optimality ratio of our policy approaches $1$.
>
> | Variable | 100 | 200 | 300 | 400 | 500 | 600 | 700 | 800 | 900 | 1000 | 2000 | 3000 | 4000 | 5000 |
> | --- | --- | --- | --- | --- | --- | --- | --- | --- | --- | --- | --- | --- | --- | --- |
> | ID policy (our work) | 0.9261 | 0.9501 | 0.9617 | 0.9666 | 0.9706 | 0.9734 | 0.9745 | 0.9763 | 0.9776 | 0.9787 | 0.9856 | 0.9881 | 0.9898 | 0.9910 |
> | Confidence interval width | 0.0005 | 0.0003 | 0.0004 | 0.0002 | 0.0002 | 0.0002 | 0.0002 | 0.0002 | 0.0002 | 0.0001 | 0.0002 | 0.0001 | 0.0001 | 0.0001 |
>
> Due to the limited time of the rebuttal period, we have only performed simulations on a small number of instances. We could add a few more simulations and include the results in our final version if our paper gets accepted.
>
> > Question 1: Do you have any insights on how results extend to the case in which we consider continuous sub-MDPs?
> >
>
> We assume that the reviewer is referring to the continuous-space sub-MDPs. This is an interesting point, and we do think that our algorithm and analysis can possibly be extended to continuous spaces. To see this, observe that the real-time phase of the ID policy only needs to sample from the optimal single-armed policies $\bar{\pi}\_{i}^{\ast}$’s, which is possible even when the space is continuous.  Moreover, our optimality gap does not explicitly depend on the sizes of state and action spaces, as can be seen from the expression of $C\_{\text{ID}}$ in second paragraph below Theorem 1.  Therefore, if we can learn the optimal single-armed policies $\bar{\pi}\_{i}^{\ast}$ in the continuous sub-MDPs, our results can possibly carry through to continuous spaces.
>
> > Question 2: What are the main technical challenges in considering learning policies in unknown sub-MDPs? For instance, is would be possible to apply concepts by constrained reinforcement learning?
> >
>
> As mentioned at the beginning of this rebuttal, a reasonable approach for learning a policy for the N-armed WCMDP could be learning the model parameters and then applying the ID policy based on the learned parameters.  To do this, we first learn the model parameters and construct the optimal single-armed policies for each sub-MDP.  As the reviewer pointed out, this can be done through approaches for constrained RL.  The main challenge in this approach is to handle how the errors in each sub-MDP affect the N-armed systems.  Note that actions generated by single-armed policies may not be feasible in the N-armed system since they may not satisfy the hard constraints (this is where the ID policy steps in).  Therefore, errors in each sub-MDP may have a non-linear effect and characterizing this effect is highly non-trivial.
>
> > Question 3: In the paragraph at lines 147--158, is it really necessary to indicate the policy when considering the vectors of MDPs states and actions?
> >
>
> We thank the reviewer for the pointing out this issue. We indicate the policy $\pi$ in superscripts mainly because $\pi$ is the decision variable of the optimization problem formulated in (3), and we want to emphasize the dependency on the decision variable. We can also omit $\pi$ in the paragraph at lines 147--158 if that is more natural.
>
> > Question 4: Is $R^*(N, \mathbf{S}\_0)$ only the optimal reward, or is it accounting for constraint satisfaction too?
> >
>
> It is the optimal reward; we consider the hard constraints and do not allow constraint violations.
>
> > Question 5: “In Lemma 1, what happens to constraint violation when employing the solution of the relaxed problem?”
> >
>
> Employing solutions of the relaxed problem would cause the constraint to be only satisfied in the long-run average sense, since the relaxed constraint (5b) is in terms of the long-run average state-action frequency $y(s,a)$. This is why we need Algorithms 1 and 2: we want to turn the solution of the relaxed problem into a policy that satisfies the hard constraint in (3b).

---

> > ### Comment · Reviewer_oixq · 2025-08-03
> >
> > I thank the authors for the detailed answers, especially for having run the toy experiment, which I suggest to include in the manuscript for enforcing the main claims. All my concerns have been addressed. I have increased the rate to 5.

---

### Official Review · Reviewer_NG7h · 2025-07-02

**Clarity:** 3
**Significance:** 2
**Originality:** 3
**Rating:** 5
**Confidence:** 3

**Summary:**

This paper studies the problem of fully heterogeneous, average-reward, budget-constrained weakly-coupled MDP (WCMDP) under the setting of planning (transition kernels and reward functions are known). For the first time, they design the algorithm of $\textit{ID policy with reassignment}$ and prove it achieves an $O(\sqrt{N})$ optimality gap as the number of arms $N$ grows to infinity. The authors novelly design a clean projection-based Lyapunov function analysis that bypasses the need for symmetry or state aggregation. But this paper comes without any experiments.

**Questions:**

please refer to weakness

**Ethical Concerns:**

["NO or VERY MINOR ethics concerns only"]

**Final Justification:**

I think this work is strong in its theoretical contribution, providing a novel technique. So I will keep my positive rating.

**Limitations:**

I didn't find the discussion about the limitations and societal impact. Since this work is theoretical in nature, it is acceptable for that.

**Quality:**

3

**Strengths And Weaknesses:**

Strength:
1. The presentation is good. Writing is easy to follow.
2. For the first time, this paper closes the gap and provides the first asymptotic optimality result for average-reward, hard-constrained WCMDP. Also this paper is technically solid.
3. Technically, they provide a novel construction of a Lyapunov function based on projected features across heterogeneous arms.

Weakness:
1. I am not convinced by the problem setting's practicality. The average reward setting is not widely used since it is not easy to learn, and not as stable as the discounted reward setting. In this case, the authors should provide at least a synthetic toy case. Second, the hard instantaneous cost constraint is fixed for each time. It might be more practical to consider the cumulated cost for a period of time?
2. It will be better to provide more comparisons on the techniques with the discounted WCMDP literature, since the authors mentioned the optimality result is provided in that case.

---

> ### Author Rebuttal · Authors · 2025-07-30
>
> > Weakness 1.1: I am not convinced by the problem setting's practicality. The average reward setting is not widely used since it is not easy to learn, and not as stable as the discounted reward setting.  In this case, the authors should provide at least a synthetic toy case.
> >
>
> Thanks for bringing up this point. While learning in average-reward MDPs is harder than in discounted-reward MDPs, it is an active research area and increasingly efficient learning algorithms have been proposed. See, for example, the following paper for a review of the recent developments in learning average-reward MDPs:
>
>     Lee, Jongmin, Mario Bravo, and Roberto Cominetti. "Near-optimal sample complexity for MDPs via anchoring." *arXiv preprint arXiv:2502.04477* (2025).
>
> For weakly-coupled MDPs, the policy that we propose can serve as a good baseline policy.  We are currently working on leveraging such a near-optimal planning policy to develop learning algorithms with a lower sample complexity.
>
> As for the experiments, we have performed simulations on some randomly-generated WCMDP instances, whose setup and data can be found in our reply to the Reviewer oixq. Due to the limited time of the rebuttal period, we have only performed simulations on a small number of instances. We could add a few more simulations and include the results in the final version if our paper gets accepted.
>
> > Weakness 1.2: Second, the hard instantaneous cost constraint is fixed for each time. It might be more practical to consider the cumulated cost for a period of time?
> >
>
> We agree that the cumulated cost is an interesting setting.  That said, the hard instantaneous cost constraint is a natural model for allocating perishable resources, such the computing time of the servers, the availability of nursing personnels, the network bandwidth, etc.
>
> > Weakness 2: It will be better to provide more comparisons on the techniques with the discounted WCMDP literature, since the authors mentioned the optimality result is provided in that case.
> >
>
> Thanks for the suggestion. We have briefly discussed the difference between the existing results for discounted-reward WCMDPs and average-reward WCMDPs at the end of Appendix B. We could expand on that to comment more on the techniques.
>
> At a high-level, the main challenge in the discounted setting is achieving *transient* optimality. To show that a certain policy achieves asymptotic optimality, one needs to show that the stochastic sample paths under the policy concentrate around an “optimal” deterministic path [See, e.g., 17, 52]. The concentration of sample paths usually follow from the concentration of *per-step* transitions, when the number of arms is large.
>
> In contrast, in the averge-reward setting, the challenge mainly lies in understanding the long-term behavior of the system, and one needs to show that the steady-state distribution of the system state has some concentration properties. Such concentration properties do directly follow from the concentration of per-step transitions; intuitively, some global stability is needed to prevent the stochastic noise from accumulating over the infinite time. Establishing such global stability was a long-standing challenge resolved only recently [22,25,26,47], and a natural technique is to use Lyapunov functions.

---

> > ### Comment · Reviewer_NG7h · 2025-08-03
> >
> > I thank the authors for addressing my concerns. All my questions are addressed, and I will keep my positive evaluation for this work.

---

### Official Review · Reviewer_aSGL · 2025-07-13

**Clarity:** 4
**Significance:** 4
**Originality:** 3
**Rating:** 5
**Confidence:** 3

**Summary:**

The paper considers a class of problems known as weakly coupled Markov Decision Processes. In this framework, the linkage between individual components, or arms, is through joint cost constraints, while each arm operates as its own independent MDP. The objective is to solve for the optimal policy for the entire system, keeping these shared constraints in mind. Typically, the state space for the entire system is exponential in the number of arms, making the problem difficult to solve directly. To approach this, the paper considers the Linear Programming relaxation of the original MDP. The best policy obtained from this LP relaxation provides an upper bound on performance, since the cost constraint is relaxed to only require satisfying a time-averaged version of the constraint. The authors then propose an "ID reassignment" based policy to determine the optimal arm-pulling strategy, such that the optimal stationary distribution obtained from the LP optimization problem is eventually tracked. They analyze the regret associated with the LP optimal solution, which serves as an upper bound on the actual regret with respect to the original MDP. The rest of the paper is dedicated to proving that the ID assignment indeed tracks the LP optimal policy in an asymptotic sense. The technical novelty lies in the proof technique, where the a unique Lyapunov function is synthesized for this fully heterogenous setting.

**Questions:**

1. In equation 8, is there a reason for choosing half the budget per cost constraint as the limit to determine the active constraints? Or is it an arbitrary choice?
2. In equation 7, is there any intuition beyond setting the policy to be uniform in transient states?
3. Does line 259 imply that as the arms with lower indices approach their stationary values, the cost constraints are satisfied in a sense that lets more arms follow their LP-optimal policies?
4. It will help to include the intuition behind Algorithm 1 in the main body of the paper since this is the main algorithmic contribution of the paper.
5. Is there any intuition behind choosing the cost and reward vectors as the feature vectors?
6. Is the regret quantified with respect to the best stationary optimal policies or best optimal policies in a wider class of policies (as initially pointed at)?

Minor comment: In the beginning the authors mention that lim inf and lim sup of the average cost are the same in the finite state action setting. However, this is true subject to certain constraints on the Markov chain induced by the policy. Unichain and aperiodicity are sufficient (as assumed later) but this assumption needs to be made clearer earlier on.

**Ethical Concerns:**

["NO or VERY MINOR ethics concerns only"]

**Final Justification:**

The authors have addressed most of my concerns satisfactorily and hence, I am increasing my score.

**Limitations:**

Yes.

**Paper Formatting Concerns:**

None. Paper is incredibly well written.

**Quality:**

4

**Strengths And Weaknesses:**

Strengths:

1.  The paper's primary claim is that it provides the first asymptotic optimality result for fully heterogeneous average-reward WCMDPs. This is a significant step beyond existing work, which has focused on homogeneous or simpler typed heterogeneous settings.
2. The design of a novel Lyapunov function is a theoretical contribution which can be leveraged as a tool to analyze other heterogeneous systems.

Weaknesses:

1. A more complete work would characterize the performance of the ID-reassignment policy in a finite time sense. The paper presents the asymptotic performance where the performance is characterized over the infinite time horizon.
2. As the number of arms $N$ tends to infinity, the assumption on the finiteness of mixing time may fall through.
3. It is unclear as to what the approach should be if the LP relaxation has multiple optimal solutions.
4. $C_{ID}$ seems to scale really badly with the problem parameters, making the result vacuous in large complex settings.

---

> ### Author Rebuttal · Authors · 2025-07-30
>
> > Weakness 1: did not characterize the finite-time performance
> >
>
> Thanks for bringing up this point. In fact, our proofs of Theorem 1 can be easily adapted to give the following finite-time performance bound:
>
> $$
> R^*(N, \mathbf{S}\_0) - \frac{1}{T} \sum\_{t=0}^{T-1} (\text{expected reward per arm at time $t$}) \leq \frac{C\_{\text{ID}}}{\sqrt{N}} + \frac{2r\_{\max} + L\_h}{\rho\_V T}
> $$
>
> To prove this, we will only need to slightly modify Lemma 4 and 5. Note that the Lemma 5 actually gives a finite-time bound on the expectation of the Lyapunov function $V(\mathbf{X}\_t)$ (The equation above (82)), and Lemma 4 can be modified to show that:
>
> $$
> R^*(N, \mathbf{S}\_0) - \frac{1}{T} \sum\_{t=0}^{T-1} (\text{expected reward per arm at time $t$}) \leq \left(1 + \frac{2r\_{\max}}{L\_h}\right) \frac{1}{T N}\sum\_{t=0}^{T-1} \mathbb{E}[V(\mathbf{X}\_t)] + \frac{2r\_{\max}K\_{\text{conf}}}{\sqrt{N}}
> $$
>
> Combining these two inequalities would give the finite-time performance bound that we claimed at the beginning.
>
> We will add this result and its derivation to the paper.
>
> > Weakness 2: as $N\to\infty$, the assumption on the finiteness of mixing time may fall through.
> >
>
> Our theorem 1 currently assumes the existence of a constant-order upper bound $\tau$ on the mixing times of all arms (Assumption 1). While this assumption might sound strong, there are some straightforward ways to relax this assumption:
>
> - First, if an $O(1/\sqrt{N})$ fraction of arms have unbounded mixing times, we can modify the algorithm to ignore these arms, which only causes $O(1/\sqrt{N})$ degradations in the optimality gap and does not affect the order of our bounds. Moreover, if we only aim for asymptotic optimality, i.e. $o(1)$ optimality gap, we can have at most $o(1)$ fraction of arms with unbounded mixing times.
> - Second, while we mainly focus on the scaling of $N$, Theorem 1 remains true even if the mixing time $\tau$ scales up with $N$. In that case, Theorem 1 implies that the optimality gap of our policy is at most $O(\tau^4 / \sqrt{N})$  (see the second paragraph below Theorem 1), so asymptotically optimality is still achieved when $\tau = o(N^{1/8})$.
>
> We could comment on these two relaxations of Assumption 1 in the paper.
>
> > Weakness 3: It is unclear as to what the approach should be if the LP relaxation has multiple optimal solutions.
> >
>
> We would like to point out that our policy and proofs do not rely on the uniqueness of the optimal solution to the LP relaxation. In the presence of multiple optimal solutions, any of them satisfying Assumption 1 can be used to define an instance of the ID policy, and Theorem 1 applies to all such instances. We can clarify this point in Section 3.
>
> > Weakness 4: $C\_{\text{ID}}$ seems to scale badly, making the result vacuous in large complex settings
> >
>
> We acknowledge that our analysis focuses on the scaling with $N$ and does not aim to optimize the dependence of the optimality gap bound on other parameters.
>
> That said, we would like to highlight an interesting fact: $C\_{\text{ID}}$ does not explicitly depend on the sizes of the state or action spaces. This implies that our bound in Theorem 1 could still be informative for some WCMDP instances with large state-action spaces and many arms, provided that the relevant quantities (such as the mixing time bound $\tau$) remain moderate.
>
> > Question 1: In equation 8, is there a reason for choosing half the budget per cost constraint as the limit to determine the active constraints? Or is it an arbitrary choice?
> >
>
> Thanks for the question. This choice is arbitrary; any constant fraction strictly between $0$ and $1$ would yield the same $O(1/\sqrt{N})$ optimality gap. We will clarify this point in our revision.
>
> > Question 2: In equation 7, is there any intuition beyond setting the policy to be uniform in transient states?
> >
>
> Thanks for the question. To clarify, there is no unique “correct” way to define the single‑armed policy $\bar{\pi}\_{i}^{\ast}$ in transient states. For the purpose of proving Theorem 1, any definition of the transient behavior of $\bar{\pi}\_{i}^{\ast}$ suffices, provided it ensures the induced Markov chain is an aperiodic unichain with mixing time bounded by $\tau$ (Assumption 1). However, the specific choice may affect the required strength of Assumption 1.
>
> The rationale behind our definition of $\bar{\pi}\_{i}^{\ast}$ is that it maximizes the "connectivity" of the induced chain. Specifically, if any definition of $\bar{\pi}\_{i}^{\ast}$ in transient states induces a unichain for arm $i$, then this uniform‑action definition also induces a unichain, since it assigns positive probability to every action.
>
>
>
> > Question 3: Does line 259 imply that as the arms with lower indices approach their stationary values, the cost constraints are satisfied in a sense that lets more arms follow their LP-optimal policies?
> >
>
> Your understanding is accurate. We will make this point more explicitly in our revision.
>
> > Question 4: It will help to include the intuition behind Algorithm 1 in the main body of the paper.
> >
>
> We thank the reviewer for the suggestions. We have provided some intuitions on Lines 263—269, but we could also move up some content in Appendix C to the main body.
>
> > Question 5: Is there any intuition behind choosing the cost and reward vectors as the feature vectors?
> >
>
> We design the feature vectors to preserve sufficient information of the system state $\mathbf{X}\_t$ so that, after projection, we can still determine (i) whether a subset of arms can follow the optimal single-armed policies, and (ii) the instantaneous reward of the subset of arms if they follow the optimal single-armed policies. The first goal requires us to include the cost vector so that we can check if the constraints are satisfied; the latter requires us to include the reward vector.
>
> > Question 6: Is the regret quantified with respect to the best stationary optimal policies or best optimal policies in a wider class of policies (as initially pointed at)?
> >
>
> Thanks for pointing out this confusion. We would like to clarify that these two ways of quantifying the regret are equivalent in our setting. Note that the WCMDP problem is an MDP with finite state and action spaces, so, by Theorem 9.1.8 of [35], there exists a stationary Markov policy that is optimal within the class of history-dependent randomized policies (the widest possible class). We will revise Lines 183-185 to clarify this.
>
> > Minor comment: In the beginning the authors mention that lim inf and lim sup of the average cost are the same in the finite state action setting. However, this is true subject to certain constraints on the Markov chain induced by the policy. Unichain and aperiodicity are sufficient (as assumed later) but this assumption needs to be made clearer earlier on.
> >
>
> We would like to clarify that we do not need the additional assumptions that the reviewer mentions. There are about two places where we mention that lim inf equals lim sup, which we clarify one-by-one:
>
> - In the paragraph below (3), we are only arguing that the *optimal* limsup average reward equals the *optimal* liminf average reward, which always hold in the finite state-action setting, by Proposition 9.1.6 of [35] cited in this paragraph.
> - At the top of Page 5, we argue that the limsup average reward equals the liminf average reward for a finite-state Markov chain. Note that we fix the initial state $\mathbf{S}\_0$ and consider the long-run average of the expectation, so we do not really need unichain or aperiodicity.

---

> > ### Comment · Reviewer_aSGL · 2025-08-09
> >
> > Thanks for addressing most of my concerns in detail. I have increased my score.

---

> ### Author Response · Authors · 2025-08-08
>
> Dear Reviewer,
>
> Thanks again for reviewing our paper and providing feedback.  As the Author-Reviewer discussion phase is concluding soon, we would like to confirm whether we have fully addressed the questions you raised.
>
> --Authors

---

### Decision · Program_Chairs · 2025-09-17

**Decision:**

Accept (spotlight)

**Comment:**

Reviewers agree on the significance of the theoretical results, the methodological novelty, and the remarkable quality of the writing.
I encourage the authors to implement the proposed improvements in the final version and to include a polished version of the experiments that were presented in the rebuttal. The new finite-time bounds should also be added.